# The Development of Intuitive and Analytic Thinking in Autism: The Case of Cognitive Reflection

**DOI:** 10.3390/jintelligence11060124

**Published:** 2023-06-20

**Authors:** Kinga Morsanyi, Jayne Hamilton

**Affiliations:** 1Department of Mathematics Education, Loughborough University, Loughborough LE11 3TU, UK; 2School of Psychology, Queen’s University Belfast, Belfast BT7 1NN, UK; jayne@studyseed.co.uk

**Keywords:** analytic thinking, attention to detail, autism, cognitive reflection, developmental change, intuition, reasoning

## Abstract

The cognitive reflection test (CRT) is a short measure of a person’s ability to resist intuitive response tendencies, and to produce normatively correct responses that are assumed to be based on effortful, analytic thinking. A remarkable characteristic of the CRT is that although the questions are open-ended, for each item, the vast majority of people either produce a correct, analytic response or a typical incorrect (i.e., intuitive) response. This unique feature of the CRT makes it possible to investigate the question of whether autistic and neurotypical people share the same intuitions. We report a study that included adolescents and young adults. In both age groups, autistic and neurotypical participants were matched on age, gender, cognitive ability, and educational background. In line with previous findings, the results showed an age-related increase in analytic responding on the CRT, and a decrease in intuitive responding. Crucially, the proportion of both intuitive and analytic responses across autistic and neurotypical participants was identical in both age groups. The current results are in contrast with claims that autistic individuals have an increased tendency toward an analytic/rational type of processing, which is commonly attributed to an impairment within their intuitive reasoning mechanisms.

## 1. Introduction

Autism Spectrum Disorder (ASD) is characterized by a combination of persistent deficits in social communication and social interaction, and restricted, repetitive patterns of behavior, interests, and activities. For an individual to obtain a diagnosis of ASD, these symptoms must be present in early childhood, should cause clinically significant impairments in social, occupational, or other areas of functioning, and they should not be better explained by intellectual disability ([2]). It is estimated that the prevalence of ASD is about 1 in 36 individuals ([33]), although diagnosis rates are lower, especially in the case of women (e.g., [24]).

Individuals with ASD are often described as having a strong focus on literal meaning and explicitly presented details (e.g., [22]; [25]), and, at the same time, they might miss the “big picture” (e.g., [20]) or interpret information in a different way than other people. This may be evident in the verbal domain, for example in understanding figurative language, such as metaphors (see [26] for a systematic review and meta-analysis), although such differences may be smaller or may not even be present when autistic and neurotypical participants are carefully matched on intelligence or verbal ability, or when samples involve high-ability participants ([42], [43]).

Studies on thinking, reasoning, and judgment in autism suggested that there might be a tendency in individuals with autism to be hyper-systematic in their thinking (e.g., [3]), or to exhibit enhanced logical consistency ([16]). This has been contrasted with a tendency to lack some basic intuitions, such as the ability to read others’ emotions (e.g., [5]). Previous studies showed that autistic individuals were less susceptible to framing effects (i.e., framing gambles of equal expected values as loss/gain; [16]; [56], although see [31]), and to the conjunction fallacy ([40]). Additionally, using a self-report questionnaire, the rational-experiential inventory ([46]), several studies ([10]; [31]; [58]) showed lower levels of self-reported engagement in experiential/intuitive thinking in the ASD group, although the groups did not differ in their self-reported rational/analytic thinking styles. Additionally, [9] ([9]) found that autistic adolescents showed a reduced tendency to “jump to conclusions” (i.e., they requested more information before making a judgment about probabilistic sampling distributions), which suggests that they are less likely to follow their gut instincts than neurotypical adolescents. 

Although the above studies suggest that there may be differences between autistic and neurotypical individuals in terms of their reasoning, decision-making and judgments, there is some difficulty with interpreting the results from these studies. Specifically, most reasoning and judgment problems that contrast intuitive and normative/analytic reasoning do not make it possible to distinguish between the hypotheses that autistic participants tend to perform better on some of these problems because they have a stronger tendency to rely on analytic reasoning or because they do not share the typical intuitions of other individuals. (Of course, a third possibility is that both of these hypotheses are correct.) 

The studies based on self-report measures (e.g., [31]) suggest that the difference may only lie in intuitive reasoning and there may be no difference between autistic and neurotypical people in their analytic thinking abilities. In line with these studies, [40] ([40]) also found that although autistic adolescents were less susceptible to the conjunction fallacy (which is based on intuitive thinking), they were not any more likely to rely on the conjunction rule (i.e., analytic processing) in their judgments than neurotypical participants. Nevertheless, the evidence for a reduced tendency for intuitive thinking in autism remains scarce. 

In the current study, we investigated intuitive and analytic processing in autism, using the cognitive reflection test (CRT; [19]). The CRT measures a person’s ability to resist intuitive response tendencies, and to produce normatively correct responses that are assumed to be based on effortful, analytic reasoning. For example, consider the following item: *“If it takes 5 min for five machines to make five widgets, how long would it take for 100 machines to make 100 widgets?”* The correct answer is 5 min, but the majority of participants (about 60%—see [49]), give the incorrect, but intuitively compelling response of 100 min. Indeed, it is a very remarkable property of the CRT that, although the questions are open-ended, for each item the majority of people either produce a normatively correct/analytic response or a typical incorrect (i.e., intuitive response). In other words, using the CRT makes it possible to investigate the tendency to give intuitive vs. analytic responses relatively independently, as participants who do not give the analytic response could still produce a response that is different from the typical intuitive response. 

Performance on the CRT in autism has already been investigated in previous studies, although the results were somewhat mixed. [10] ([10]) administered the CRT to autistic and neurotypical late adolescents who were matched on age and gender (all participants in this study were males). Autistic participants gave significantly fewer intuitive and significantly more analytic responses to the CRT problems.

In another study, [11] ([11]) administered the CRT to autistic and neurotypical late adolescents and young adults. The groups were matched on gender and non-verbal intelligence, but the autistic group was slightly older. The results showed no significant difference in analytic responses to the CRT problems, but autistic participants gave significantly fewer intuitive responses (with a medium effect size). Additionally, [11] ([11]) reported that autistic traits (as measured by the Autism Spectrum Quotient—[4]) showed a small, but significant positive correlation with giving analytic responses to the CRT. In this correlational analysis, the autistic and neurotypical samples were pooled, and the effects of age, gender and diagnostic group membership were controlled.

[8] ([8]) also investigated performance on the CRT among autistic and neurotypical late adolescents. The groups were matched on age, but they were not matched on cognitive ability. Participants performed the CRT either under the typical, untimed condition or under time pressure. The results showed that (in both conditions) autistic participants produced more analytic and fewer intuitive responses than the neurotypical participants. Additionally, the effect of time pressure was similar in both groups, with a higher proportion of intuitive and lower proportion of analytic responses under time pressure. Similar to [11] ([11]), [8] ([8]) also reported that there was a small in size but significant positive correlation between autistic traits and providing correct responses to the CRT. In this analysis, they pooled the autistic and neurotypical samples, and controlled for the effect of condition (time pressure/untimed), gender, age and diagnostic status. Autistic traits were measured using the Subthreshold Autism Trait Questionnaire (SATQ; [27]).

Overall, these results seem to paint a relatively clear picture regarding the performance of autistic participants on the CRT. In all of these studies, autistic participants gave fewer intuitive responses to the CRT, and in two out of the three studies ([10]; [8]), they also gave a higher proportion of analytic responses. Two of these studies ([11]; [8]) also showed a significant positive correlation between autistic traits and the proportion of analytic responses on the CRT, although this relationship was weak. 

Nevertheless, there are some methodological concerns regarding these findings. Specifically, the studies that showed a significant difference in analytic responding between autistic and neurotypical participants did not match the samples on cognitive ability. This is problematic, because [19] ([19]) described the CRT as a ‘simple measure of one type of cognitive ability’ and reported reliable relationships between cognitive reflection and measures of intelligence (see also [45] for a meta-analysis). A recent study ([58]; Study 4) overcame this methodological issue by comparing the performance of large samples of autistic and neurotypical participants, who were closely matched on age, gender, and cognitive ability. The results showed no group difference in either analytic or intuitive responding on the CRT, which was in contract with [11] ([11]) who reported a significant difference in intuitive responding on the CRT even when the autistic and neurotypical groups were matched on cognitive ability. 

[58] ([58]; Studies 1–3) also investigated the relations between autistic traits (as measured by the AQ scale; [4]) and performance on different versions of the CRT in large online samples that were drawn from the general population. These studies showed no relationship between autistic traits in the general population and either analytic or intuitive responding on the CRT. Although in Study 3, AQ scores showed a significant (weak, positive) correlation with analytic responding on the CRT and a significant (weak, negative) correlation with intuitive responding, these correlations became non-significant when the effects of age, gender and cognitive ability were controlled for. Overall, [58]’s ([58]) findings suggest that performance on the CRT may not be related to either a diagnosis of autism or autistic traits in the general population when the effects of cognitive ability are taken into account.

In addition to ignoring the effects of cognitive ability, another potential issue with the studies of Brosnan and colleagues is that the autistic participants were recruited from a university summer school for students on the autism spectrum, which was aimed at providing insight into university life, whereas the neurotypical group was recruited from either a general summer school ([11]; [8]) or from first year student populations ([10]). The CRT has numerical content, and, although it is more than just a mathematics test (see [12]; [32]; [45]), it relies heavily on mathematics skills and mathematics-related feelings (e.g., [41]; [50]). In fact, it has even been used as part of a numeracy test ([63]). [6] ([6]) reported that an autism diagnosis was between three and seven times more likely among mathematics undergraduates than undergraduate students in medicine, law, and social sciences. Thus, a potential explanation for any group differences on the CRT between autistic and neurotypical participants is that the autistic participants simply had better mathematics knowledge/more background in mathematics.

The aim of the current study was to replicate and extend previous work regarding cognitive reflection in autism, while also offering some methodological improvements. Specifically, we made sure that participants were well-matched on age, gender, cognitive ability, and educational background. We also wanted to investigate the question of whether a potential reason for autistic participants to produce fewer intuitive responses could be that although they were not any more analytic in their thinking than their neurotypical peers, they were less susceptible to giving the typical intuitive response. To answer this question, we analyzed the responses given to each CRT item in more detail. In particular, we were interested in whether autistic participants produced more atypical (neither analytic nor intuitive) responses to these open-ended problems. 

[11] ([11]) and [8] ([8]) reported that autistic traits were related to analytic responding on the CRT, but [58] ([58]) found no such relationship. Thus, we also revisited this question in our study. Additionally, Brosnan and colleagues did not give any details regarding the nature of this relationship. In particular, the AQ ([4]) and the SATQ ([27]) have several subscales relating to social skills, communication skills, rigidity, etc. It would be of interest to better understand which of these traits may be responsible for the tendency to provide analytic responses to the CRT. 

A final, novel aim was to investigate developmental changes in cognitive reflection (and in the tendency to produce intuitive vs. analytic responses). So far, in all existing studies on cognitive reflection in autism, the participants were either late adolescents or young adults. This raises the question of whether these results would generalize to other age groups. A typical finding in the developmental literature on reasoning biases is that intuitive responding tends to decrease and analytic reasoning tends to increase throughout adolescence and into young adulthood (e.g., [14]; [28]; [29]; [38]), and a similar tendency has been identified in the case of the CRT as well ([49]). Thus, we were interested in whether any differences in intuitive and analytic responding between the autistic and neurotypical samples (if they exist) are stable across development, or whether these gaps increase or decrease with age. If these gaps change with age, this may signal a developmental advantage or delay in the case of the autistic sample.

## 2. Method

### 2.1. Participants

The participants were adolescents and adults from Northern Ireland. In the adolescent sample, there were 28 adolescents with ASD (23 males; between the ages of 11 years 9 months and 17 years 1 month; *M* = 12.80, *SD* = 1.32) and 50 typically developing adolescents (37 males; between the ages of 11 years 7 months and 14 years 4 months; *M* = 12.97, *SD* = 0.57). A chi-square test indicated that the gender distribution of the two samples was not significantly different (*p* = .413). The participants in the autistic and neurotypical samples were recruited from the same schools. In the adult sample, there were 26 participants with ASD (23 males; between the ages of 18 and 41 years; *M* = 22.81, *SD* = 6.41) and 27 neurotypical adults (22 males; between the ages of 18 and 52 years; *M* = 22.26, *SD* = 7.02). A chi-square test indicated that the gender distribution of the two samples was not significantly different (*p* = .478). All participants were recruited from university courses either at Queen’s University Belfast or Ulster University. The participants with ASD were attending various courses, including math, physics, engineering, computing, law, arts, and biological and life sciences. The participants in the neurotypical group were recruited from the same university courses to control for the potential effects of educational background and, in particular, interest and experience in mathematics. In the adolescent sample, the participants’ school shared any official diagnosis (autism, dyslexia, etc.) that the participants had as well as further details (e.g., date of diagnosis, and the person providing the diagnosis) when available. In the adult sample, the participants confirmed their own diagnosis and the details that they knew. In both samples, the participants had no diagnosis other than ASD. Table 1 presents more details regarding the characteristics of our participants. Blocked Operation Span scores were missing for two autistic participants due to a software error.

### 2.2. Materials 

The *CRT-Long* ([49]), an extended, 6-item version of the CRT ([19]) was used to measure cognitive reflection (the full list of items and their scoring can be found in Appendix A). The CRT-Long includes 3 new items beside the original items, and it has been developed with the aim of making it possible to measure cognitive reflection in lower ability samples as well (such as developmental samples), who often score zero on the original CRT. All CRT items are open-ended and there is no time limit for the participants to complete the questions. Participants’ responses were marked as analytic (i.e., correct), intuitive (which corresponds to the typical incorrect response) or other. In the current sample, Cronbach’s alpha for the scale (based on the total number of analytic responses) was 0.78. Performance on the original CRT items and the CRT-Long showed a very strong correlation (*r*(129) = 0.91 *p* < .001), confirming that the CRT-Long measured the same construct as the original CRT.

#### 2.2.1. Ability Measures Used for Matching the Adolescent Samples

To match the groups on *general intelligence*, we used a short form of the *Wechsler Intelligence Scale for Children* ([62]), consisting of the Block Design and Vocabulary subtests. This short form of the WISC is reported to have the highest reliability and validity compared to any other two-subtest short forms (see [55]). Using these scores, it was also possible to calculate the participants’ estimated IQ (see [54]). 

As a measure of *fluid intelligence*, we used Set 1 of Raven’s Advanced Progressive Matrices ([52]) consisting of 12 items. The test was administered together with 3 practice items taken from the Raven Standard Progressive Matrices ([51]). This set is usually used as a practice and screening set for the full Raven’s test as it covers all the intellectual processes assessed by the full test. However, it does not extend to the highest complexity levels. [15] ([15]) reported that Set-1 of the APM was a reliable test to be used in a short time frame as it cuts around 60% of testing time compared to the full Raven’s test, thus helping to prevent the effects of fatigue and boredom on results. 

#### 2.2.2. Ability Measures Used for Matching the Adult Samples

As a measure of *verbal intelligence*, we used the Vocabulary Subtest of the Wechsler Abbreviated Scale of Intelligence (Version Two; [61]). In the vocabulary subtest, participants have to define words which are presented to them verbally, one by one. This subtest is considered to measure expressive vocabulary, verbal knowledge, and crystallized and general intelligence. Performance also draws heavily on memory, learning ability, and concept and language development ([54]).

As a measure of *verbal working memory*, we used the Operation Span test ([18]). In this computer-based task, participants are asked to memorize a series of letters, while being distracted by math sums. The participant first completes a simple mental arithmetic problem, then they see a letter, then another arithmetic problem and another letter. This process is repeated between three and seven times, although the trial length is random and cannot be predicted. At the end of each trial, the participant has to recall the letters in the correct order. Scores are calculated by adding up the number of correct letters that were recalled in the correct order ([60]). The absolute score is based on the total number of correctly recalled letters (based on only those sequences for which all letters in the sequence were recalled in the correct order).

*Autism-Spectrum Quotient:* The 50-item AQ scale ([4]) measures a variety of autistic traits, using a four-point scale that ranges from *definitely agree* to *definitely disagree*. We used the dichotomous scoring system of Baron-Cohen and colleagues, where participants were given 1 point if they indicated that they agreed/definitely agreed with an item measuring autistic characteristics, and they were given 0 point if they indicated that they (definitely) disagreed. The scale assesses 5 different areas (with 10 questions each): *social skill (e.g.,* “*I prefer to do things with others rather than on my own”)*; *attention switching (“I frequently get so strongly absorbed in one thing that I lose sight of other things”*); *attention to detail (“I often notice small sounds when others do not”*); *communication (“I enjoy social chit-chat”*) and *imagination (“I find making up stories easy”)*. Approximately half the items are worded to produce a ‘disagree’ response, and half an ‘agree’ response, in a high scoring autistic person. Cronbach’s alpha for the full scale was 0.92 (0.83, 0.73, 0.55, 0.84, and 0.70 for the social skills, attention switching, attention to detail, communication, and imagination subscales, respectively). Higher scores on subscales indicate higher levels of autistic traits (i.e., worse social skills, attention switching, communication skills, and imagination, but better attention to detail).

### 2.3. Procedure 

The study received ethics approval from the Faculty’s Human Ethics Committee. In the case of the adolescent sample, after receiving permission from the schools to carry out the study, parents/caregivers provided informed consent and the participants provided assent. In the young adult sample, the participants were recruited from local universities. Adolescents were involved in two testing sessions. In the individual session, the participants completed the vocabulary and block design tests, which were administered by one of the experimenters. The CRT-Long and Raven’s Matrices were completed in a paper-and-pencil format in a group session (which took place on a separate day), with groups of 3–4 children, but with all participants working individually. Adults completed a paper-based version of the CRT-Long and the AQ scale, and a computer-based verbal working memory test. As part of the same session, they also completed the vocabulary test, which was administered by one of the experimenters.

## 3. Results

The data that support the findings of this study are available at (https://osf.io/t7yn4/?view_only=d7be657e7a56445897a8ccc336e44c5f) Our first analysis investigated whether participants in each sample mostly gave either the intuitive or the analytic response to the CRT-L items (the exact proportion of intuitive and analytic responses given to each item by each group is presented in Table 2). The overwhelming majority of responses to each item were either intuitive or analytic. When we looked at the autistic and neurotypical groups separately within each age group, the proportion of intuitive/analytic responses to each CRT item ranged from 72% to 100%. We also checked whether there were any other typical responses apart from the intuitive and analytic responses given to any item by either group. For most items, the responses that were neither intuitive nor analytic were only given by a very small number of participants (i.e., no other typical responses were identified). The only exception was item 5, where 12 neurotypical adolescents (i.e., 24% of the sample) gave the response “15”. Nevertheless, even in this case, a much larger proportion of the sample (58%) gave the typical intuitive response. Thus, from these analyses we can conclude that the dominant responses given by each group to each question were either the intuitive or the analytic response. In fact, overall, 88% of the responses given by the autistic adolescents were either intuitive or analytic, and the corresponding proportions were 88% for the neurotypical adolescents, 96% for the autistic adults, and 95% for the neurotypical adult group.

In an additional analysis we also checked if the proportion of atypical (i.e., neither intuitive nor analytic) responses decreased with age. Such patterns were reported for other judgment and reasoning tasks (e.g., [37], [38]; [28]). A 2 × 2 ANOVA with age group (adolescent/adult) and diagnostic group (ASD/neurotypical) on the overall number of atypical responses showed a significant effect of age (*F*(3,127) = 9.06, *p* = .003, *η_p_*^2^ = 0.07), with a lower number of atypical responses in the case of adult participants. There was no effect of diagnostic group and no interaction between age and diagnostic group (*p*s > .81). 

Our next analysis was aimed at investigating the effects of age and diagnosis on analytic performance on the CRT-L (Figure 1). A 2 × 2 ANOVA with age group (adolescent/adult) and diagnosis (autistic/neurotypical) as between-subject factors on the number of analytic responses given to the CRT-L indicated a significant main effect of age (*F*(1,127) = 53.75, *p* < .001, *η_p_*^2^ = 0.30), but no effect of diagnosis and no age by diagnosis interaction (*p*s > 0.60).

Given that our analyses indicated no significant difference in analytic responding in either the adolescent or the adult sample, we followed up the above analyses with Bayesian *t* tests, comparing analytic responding separately in the adolescent and young adult samples. In the case of non-significant results, using a Bayesian analysis makes it possible to distinguish between inconclusive evidence, as opposed to evidence that supports the null hypothesis. A Bayes factor of above 3 indicates moderate evidence in favor of the null hypothesis over the alternative hypothesis (i.e., that there was a significant difference between the autistic and neurotypical samples in analytic responding), a Bayes factor of above 10 indicates strong evidence, and a Bayes factor of between 1 and 3 indicates anecdotal evidence. A Bayes factor of 1 indicates that the evidence in favor of the null hypothesis and the alternative hypothesis is equally strong (see, e.g., [30]).

A Bayesian *t* test in the case of the adolescent group indicated moderate evidence for no difference between the autistic and neurotypical groups in analytic responding (*BF* = 3.73). In the case of the adult sample, the results also showed moderate evidence in favor of no group differences (*BF* = 3.51).

Although our results regarding intuitive responding on the CRT-L very much mirrored the results obtained in the case of analytic responses, given the theoretical significance of these results, we also report these analyses (Figure 2). A 2 × 2 ANOVA with age group (adolescent/adult) and diagnosis (autistic/neurotypical) as between-subject factors on the number of intuitive responses given to the CRT-L indicated a significant main effect of age (*F*(1,127) = 34.38, *p* < .001, *η_p_*^2^ = 0.21), but no effect of diagnosis and no age by diagnosis interaction (*p*s > .65).

A Bayesian *t* test in the case of the adolescent group indicated moderate evidence for no difference between the autistic and neurotypical groups in intuitive responding (*BF* = 3.70). In the case of the adult sample, the results also showed moderate evidence in favor of no group differences (*BF* = 3.59).

Our next analyses investigated the correlations between participants’ scores on the AQ scale and its subscales, and their tendency to give intuitive/analytic responses to the CRT-Long (Table 3). These analyses are only presented for the adult participants, as adolescents were not administered the AQ scale. We present these correlations separately for the autistic and neurotypical samples, as we did not necessarily expect that these correlations would be the same across the samples. Overall scores on the AQ scale were not correlated with either intuitive or analytic responding on the CRT in either the autistic or the neurotypical sample. However, a very strong correlation emerged between attention to detail and analytic responding on the CRT-L in the autistic group. The correlation was in the same direction in the neurotypical group, but it was much weaker and non-significant. The other subscales of the AQ were unrelated to performance on the CRT-L, although there was some indication of a relationship between social skills and imagination and performance on the CRT-L (i.e., autistic participants who displayed better social skills and imagination—i.e., who were less “autistic” in these respects, tended to score higher on the CRT-L). 

[11] ([11]) reported that scores on the AQ scale showed a small, but significant positive correlation with giving analytic responses to the CRT when the autistic and neurotypical samples were pooled, and the effects of age, gender and diagnostic group membership were controlled. We replicated this analysis in our sample including both total scores on the AQ scale and each subscale score separately. Total scores on the AQ scale showed a very weak, non-significant, negative correlation with analytic responding on the CRT (*r*(48) = −0.08, *p* = .563). Attention to detail showed a moderate positive correlation with analytic responding on the CRT (*r*(48) = 0.46, *p* = .001). Additionally, there was a significant negative correlation between scores on the imagination subscale of the AQ scale and analytic responding on the CRT (*r*(48) = −0.32, *p* = .025). This latter finding indicated that people with lower imagination scores (i.e., who were more autistic-like in this respect) scored *lower* on the CRT.

## 4. Discussion

The aim of this study was to investigate the development of intuitive and analytic reasoning in autism, using the cognitive reflection test. In particular, we were interested in: (1) whether autistic participants generate the same typical intuitive responses to the CRT items as neurotypical individuals; (2) whether the autistic and neurotypical groups produce a similar number of intuitive and analytic responses; (3) whether developmental changes in performance on the CRT between adolescence and adulthood are similar across the diagnostic groups; and (4) whether analytic responding on the CRT is related to autistic traits. These questions are important from a theoretical perspective as autistic individuals are often described as hyper-rational in their thinking (see, e.g., [53]), and at the same time, it has been claimed that they show a reduced tendency to produce the typical intuitive responses to judgment and reasoning problems (e.g., [8]). 

Although performance on the CRT in autism has been investigated in the past ([10], [11]; [8]; [58]), these studies had some methodological weaknesses in terms of matching the autistic and neurotypical samples on some key characteristics. Only two studies ([11]; [58]) matched the samples on cognitive ability, and these studies reported conflicting findings. To address these issues, we administered the CRT to two samples of autistic and neurotypical participants who were well-matched on age, gender, cognitive ability and educational background. As the two samples were drawn from different age groups (i.e., early adolescents and young adults), we were also able to investigate developmental changes in performance on the CRT in both the autistic and neurotypical samples.

Our results showed that in both diagnostic groups, most participants gave either the typical intuitive or the analytic response to each CRT item (participants from each group did this over 85% of the time). Thus, although the problems were open-ended, autistic participants generated exactly the same types of responses as neurotypical individuals. Moreover, in line with the findings of [58] ([58]), there was no difference between the groups in the proportion of intuitive and analytic responses that they gave. This pattern was exactly the same in both the adolescent and adult samples. Thus, we replicated this result in two independent samples. A Bayesian analysis indicated moderate evidence in the case of both adolescents and young adults that there was no difference between the diagnostic groups in their tendency to give either intuitive or analytic responses to the CRT. 

With regard to developmental changes, adults produced significantly more analytic, and significantly fewer intuitive responses than adolescents (see also [49]), and the difference between age groups was exactly the same in both diagnostic groups. This suggests no difference in the development of cognitive reflection between neurotypical and autistic participants. Regarding the large developmental difference between age groups, it should be noted that the adult participants were highly educated (as they were university students), and probably not representative to a general young adult population. Thus, our results may overestimate the typical developmental differences. Nevertheless, the difference between age groups was large, and for this reason they are likely to be indicative of the actual developmental pattern. 

Following [11] ([11]), [8] ([8]), and [58] ([58]), we also investigated the relationships between autistic traits and performance on the CRT. In line with [58] ([58]), we found no relationship between overall AQ scores and cognitive reflection. Nevertheless, we found a very strong correlation between attention to detail and performance on the CRT in the autistic group. This correlation between attention to detail and cognitive reflection was also present when the diagnostic groups were pooled, and we controlled for the effects of age, gender, and diagnosis. Additionally, we found a moderate *negative* correlation between the imagination subscale of the AQ and analytic thinking on the CRT when the samples were pooled. In other words, participants with *better* imagination skills performed better on the CRT, which is in contrast with the idea that, in general, autistic traits are positively related to cognitive reflection.

In terms of the relation between attention to detail and cognitive reflection, a detail-focused processing style has been linked to systematic processing and literal interpretation of materials, which might also be coupled with a reduced tendency to process the “gist” of experiences or presented materials (cf. [35]). That is, a detail-focused processing style might make the autistic group less likely to rely on pragmatic or other contextual cues. A recent study has also found a link between attention to detail and analogical reasoning skills in an autistic sample ([43]). It would be interesting to check in future studies if attention to detail is also linked to performance on other types of judgment and reasoning problems in the case of both autistic and neurotypical samples. Investigating the relations between attention to detail and reasoning skills in autism is particularly interesting as it has been proposed that attention to detail might be linked to special skills and talents in autism (e.g., [7]; [23]). 

Overall, our findings concur with [58] ([58]) in showing that there is no difference between autistic and neurotypical individuals in cognitive reflection when the two samples are matched on cognitive ability. Moreover, a novel finding of our study is that cognitive reflection develops exactly the same way in autistic individuals as in neurotypical samples. Additionally, also in line with [58] ([58]), we found no evidence for a link between cognitive reflection and autistic traits, with the exception of a link with attention to detail. These findings fail to replicate the results of previous studies by Brosnan and colleagues ([10], [11]; [8]). The reason that Brosnan and colleagues found significant group differences may be that they did not match the autistic and neurotypical samples on key cognitive characteristics, including age, cognitive ability, and educational background. This highlights the importance of taking into account the cognitive requirements of reasoning and decision-making tasks when designing studies that compare autistic and neurotypical samples with the aim of understanding potential differences in their reasoning and decision-making skills. Studies in this field remain scarce (although see [36]). Nevertheless, methodological rigor is essential to avoid results that turn out to be false positives, and could be used to perpetuate myths and stereotypes about autism. 

In terms of the implications of the current findings, it is interesting to consider to what extent performance on the CRT can be considered as an indication of intuitive and analytic reasoning ability in general. The CRT has been an extremely popular measure, which can be attributed to the fact that it correlates moderately or strongly with a range of reasoning and decision-making tasks, as well as with measures of moral judgments and values, religious and paranormal beliefs, and creativity (see, e.g., [47] for an overview). In this respect, it can be considered an ideal measure that captures an essential skill or predisposition that is necessary for analytic thinking (which may be described as the ability to suppress tempting, but incorrect responses in favor of a more effortful, analytic response—e.g., [19]; [59]). Nevertheless, it has been questioned whether the CRT is a reliable and useful measure of intuitive thinking, or whether it purely measures analytic thinking (cf., [48]). Indeed, there is an ongoing debate in the literature whether intuitive and analytic thinking should be best conceptualized as being dependent on separable cognitive processes/systems or whether a unitary system underlies both intuitive and analytic thinking (see, e.g., [17]; [44]). Given that the correlation between CRT performance and performance on other reasoning and judgment tasks tends to be moderate, we can also expect that group differences may be (at least to some extent) task-specific. Thus, finding no group differences in cognitive reflection does not necessarily imply that autistic and neurotypical samples would not differ on other reasoning or decision-making tasks.

Even though autistic and neurotypical participants produced the same proportion of intuitive and analytic responses in our study, we have not measured the time taken to produce those responses. There may be differences between the groups in terms of the time course of producing these responses or they may not produce these responses with the same level of confidence. For example, although analogical reasoning ability has been found to be unimpaired in autism (e.g., [21]; [39]), autistic participants took longer and exerted more effort when they had to verify verbal analogies ([43]). Such differences could have an impact on reasoning performance in real-life settings, when responses need to be formulated quickly. Future studies could investigate the metacognitive monitoring and control processes (e.g., [1]) that accompany reasoning and decision-making in autism. 

[34] ([34]) found that performance on the CRT was related to metacognitive skills. In particular, analytic thinkers (who scored higher on the CRT) were less overconfident than people who scored lower, and they were better able to discriminate between whether they gave correct vs. incorrect responses, as reflected by their confidence in their responses. Their confidence ratings were also more strongly related to the objective difficulty of the problems. A potential implication of these findings is that metacognitive monitoring skills in relation to reasoning processes may be similar across autistic and neurotypical populations. Nevertheless, this needs to be confirmed empirically, and across a range of tasks.

Existing research into metacognitive skills in autism suggests that there may be differences in metacognitive accuracy between autistic and neurotypical samples, although so far this research has focused on memory performance. According to a meta-analysis ([13]), autistic participants do not differ from matched controls in terms of the accuracy of their judgements of learning (i.e., judgments about whether they will be able to recall newly learnt content). Nevertheless, metacognitive accuracy in autism was found moderately reduced on judgements-of-confidence tasks (where after each response, participants had to rate how confident they were that their answer was correct), although this difference may only be present in the case of autistic children, and not in autistic adults. Overall, research into metacognitive skills in autism remains scarce, and many studies suffer from methodological issues (cf., [13]).

In summary, our findings suggest that there is no general tendency in autism to produce a higher proportion of analytic responses, or to produce fewer intuitive responses than neurotypical individuals. This pattern was replicated in two independent samples (i.e., in adolescents and adults), and the strength of evidence for no group differences was moderate in both samples. Autistic traits in general were also unrelated to both intuitive and analytic reasoning performance, although attention to detail was positively related to analytic reasoning. This raises the possibility that, when producing analytic responses, there may be some differences in the underlying cognitive processes between autistic and neurotypical participants. As reasoning and decision making in autism are under-researched areas, it is important to carry out more research into these topics. These investigations could also involve looking into the metacognitive monitoring and control processes that accompany intuitive and analytic responding in autism. As these skills rely heavily on cognitive resources (which may include general or fluid intelligence, working memory and executive functioning skills), and may also be influenced by relevant knowledge and educational background (see, e.g., [57]), it is essential that any studies into reasoning and decision-making skills carefully match the autistic and neurotypical samples on relevant cognitive characteristics to avoid false positives. Future studies could also explore further the role of attention to detail in reasoning performance in autism.

## Figures and Tables

**Figure 1 jintelligence-11-00124-f001:**
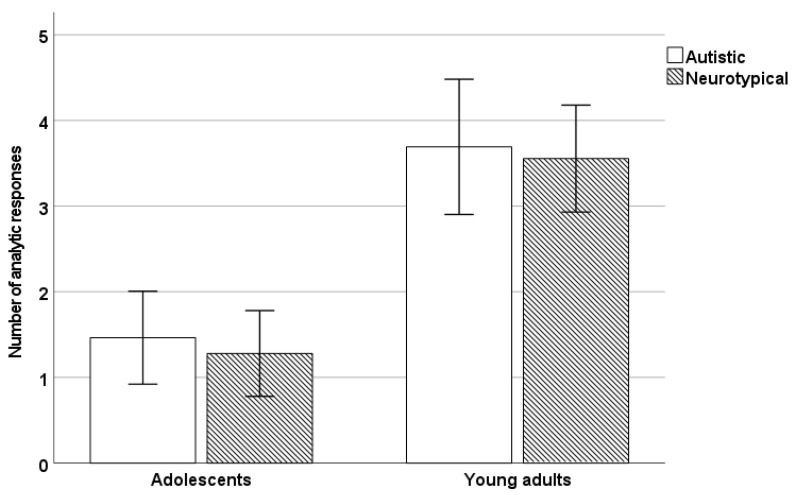
The number of analytic responses given by the autistic and neurotypical groups within each age group (error bars represent 95% CI).

**Figure 2 jintelligence-11-00124-f002:**
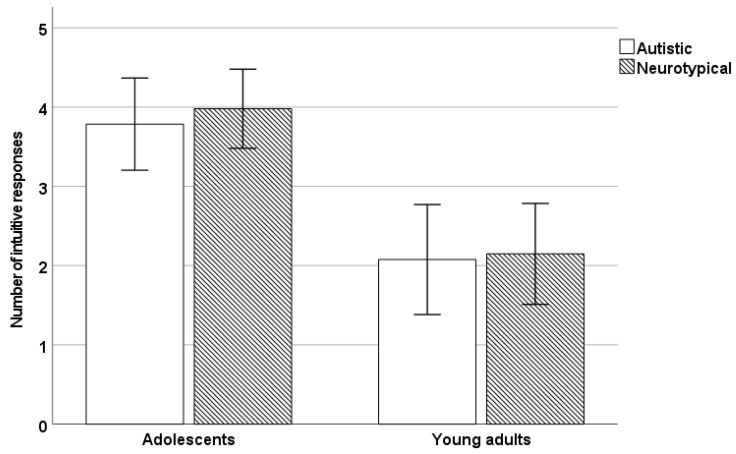
The number of intuitive responses given by the autistic and neurotypical groups within each age group (error bars represent 95% CI).

**Table 1 jintelligence-11-00124-t001:** Sample characteristics and matching (*t*-tests were run to compare the groups on each measure).

		ASD *M* (SD)	Neurotypical *M* (SD)	*p* Value
Adolescents	Age (years)	12.80 *(1.32)*	12.97 *(0.56)*	.525
	WISC-IV Vocabulary	37.68 *(7.37)*	39.16 *(5.60)*	.321
	WISC-IV Block Design	48.64 *(8.17)*	48.18 *(8.70)*	.818
	Raven’s APM	7.43 *(2.49)*	7.78 *(2.08)*	.507
	Estimated IQ	102.29 *(11.84)*	104.00 *(11.61)*	.536
Adults	Age (years)	22.81 *(6.41)*	22.26 *(5.39)*	.768
	WASI-II Vocabulary	41.08 *(8.67)*	39.19 *(8.59)*	.429
	Operation Span	14.54 *(7.37)*	14.11 *(9.09)*	.854
	AQ	30.11 *(7.77)*	16.04 *(5.37)*	<.001

**Table 2 jintelligence-11-00124-t002:** The proportion of analytic and intuitive responses given to each item split by age and diagnostic group.

	Analytic Responses				Intuitive Responses			
	Adolescents		Adults		Adolescents		Adults	
	ASD	NT	ASD	NT	ASD	NT	ASD	NT
Item 1	0.18	0.20	0.39	0.37	0.75	0.76	0.58	59
Item 2	0.32	0.28	0.61	0.44	0.50	0.60	0.39	0.52
Item 3	0.11	0.18	0.62	0.63	0.82	0.76	0.35	0.30
Item 4	0.61	0.36	0.81	0.93	0.32	0.60	0.19	0.07
Item 5	0.07	0.14	0.58	0.37	0.77	0.58	0.35	0.48
Item 6	0.18	0.12	0.69	0.81	0.61	0.68	0.23	0.19
Total	0.24	0.21	0.62	0.59	0.61	0.66	0.35	0.36

**Table 3 jintelligence-11-00124-t003:** Correlations between performance on the CRT-L and autistic traits across diagnostic groups (the correlations for neurotypical participants are presented in brackets).

	CRT-L Intuitive	CRT-L Analytic
Social skills	0.22 *(0.13)*	−.25 *(*−*0.13)*
Attention switching	−0.10 *(0.22)*	−0.002 *(*−*0.14)*
Attention to detail	−0.72 ** *(*−*0.29)*	0.72 ** *(0.21)*
Communication	0.09 *(0.001)*	−0.16 *(0.13)*
Imagination	0.29 *(*−*0.01)*	−0.31 *(0.02)*
AQ total	0.01 *(*−*0.005)*	−0.09 *(0.04)*

** *p* < .001.

## Data Availability

The data that support the findings of this study are available at https://osf.io/t7yn4/?view_only=d7be657e7a56445897a8ccc336e44c5f.

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
