# Peer review of "The Development of Intuitive and Analytic Thinking in Autism: The Case of Cognitive Reflection"

_jintelligence, 2023, doi:10.3390/jintelligence11060124_

Round 1
Reviewer 1 Report
Individuals with ASD have been described as having a strong focus on literal meaning and explicitly. Hence the CRT is adopted in this study. The authors pointed out that, although there have been reported some results (e.g., It is intuitive reasoning that characterizes the thinking of ASD), the problems of the previous studies such as non-matching of the samples on cognitive ability. This study aims to replicate the results of the previous studies regarding cognitive intuitive and analytic thinking of ASD with the matching above. In line with previous findings (Primi et al,. 2016), the results showed an age-related increase in analytic responding on the CRT, and a decrease in intuitive responding. However, their results are in contrast with claims that autistic individuals have an increased tendency toward an analytic/rational type of processing (e.g., Ro- zenkrantz et al., 2021). The results look novel and interesting. However, it is a little tough for me to follow the argument of the authors because of seeming insufficient explanations. Hence, I recommend the authors to add more detailed explanations in some parts.
1. Participants and Table 1
This study sounds that the samples of control, ASD, and NT are from both adolescences and adults. However, I cannot find the description of NT sample from adolescences. Furthermore, no statistics on NT in Table 1.
2. Table 2 and analysis
Means and SDs of the control group are now shown. Why not? Why are the data of the control group excluded from the analysis?
Minor request
L.130- 131
Please make citation about “two out of these studies” and make clear if “two of these studies(L.131)” are the same as “two out of these studies”. (“The two of these studies” in place of “Two of these studies”?)
Reviewer 2 Report
This seems to be an interesting and well-conducted study, finding no support for a link between autism and enhanced analytical thinking. The findings are supported by Bayesian analyses and there is internal replication across different measures and samples.
My main comment is that the authors have completely missed - intentionally or unintentionally - the recent research by Taylor et al. (https://psycnet.apa.org/fulltext/2022-44065-001.html), which largely finds the same pattern of results using similar methods (i.e., group matching on general cognitive ability). I would like the authors to rework their paper in careful view of the findings and methods in this recent research, including where measures and results differ (slightly).
No major comment - writing could be improved in terms of conciseness
Round 2
Reviewer 1 Report
Please accept my apology. I did not follow the correspondence between 'participants' and the results. I could do it this time.
Reviewer 2 Report
I thank the authors for addressing my comments thoroughly. Congrats on a nice paper.